# Hide and Find: A Distributed Adversarial Attack on Federated Graph Learning

## Abstract

Federated Graph Learning (FedGL) is vulnerable to malicious attacks, yet developing a truly effective and stealthy attack method remains a significant challenge. Existing attack methods suffer from low attack success rates, high computational costs, and are easily identified and smoothed by defense algorithms. To address these challenges, we propose **FedShift**, a novel two-stage "Hide and Find" distributed adversarial attack. In the first stage, before FedGL begins, we inject a learnable and hidden "shifter" into part of the training data, which subtly pushes poisoned graph representations toward a target class's decision boundary without crossing it, ensuring attack stealthiness during training. In the second stage, after FedGL is complete, we leverage the global model information and use the hidden shifter as an optimization starting point to efficiently find the adversarial perturbations. During the final attack, we aggregate these perturbations from multiple malicious clients to form the final effective adversarial sample and trigger the attack. Extensive experiments on six large-scale datasets demonstrate that our method achieves the highest attack effectiveness compared to existing advanced attack methods. In particular, our attack can effectively evade 3 mainstream robust federated learning defense algorithms and converges with a time cost reduction of over 90%, highlighting its exceptional stealthiness, robustness, and efficiency.

## 1 Introduction

Federated Graph Learning (FedGL) (He et al., 2021; 2022; Xie et al., 2023; Liu et al., 2024) , as a novel distributed learning paradigm, enables collaborative Graph Neural Networks (GNNs) (Scarselli et al., 2008) training without direct data sharing, effectively solving real-world data privacy issues (Goddard, 2017) and finding wide applications in domains such as disease prediction (Peng et al., 2022) and recommendation systems (Wu et al., 2022; Baek et al., 2023). On FedGL, data owners serve as clients, locally train models on their own private graph data, and submit only model updates to a central server for aggregation, building a shared global model while preserving data privacy (Kairouz et al., 2021).

Like ordinary graph learning, federated graph learning also faces security issues (Dai et al., 2023; Wu et al., 2024; Yang et al., 2024a). To make the model produce incorrect predictions, an attacker can control one or more malicious clients to inject various triggers (such as different subgraphs) into parts of the training data and relabel them to the attacker's predefined target label (different from their original label), in order to jointly implant the backdoor signal into the global model (Xie et al., 2019; Xi et al., 2021; Xu et al., 2024).

However, in the federated scenario, such **graph backdoor attacks** generally face two major challenges: **Challenge 1)** Malicious backdoor signals produced by malicious clients are easily smoothed out during aggregation by normal signals provided by benign clients, leading to a significant drop in attack effectiveness. **Challenge 2)** To resist signal smoothing, a direct approach is to increase the attack budget. However, larger-scale data poisoning leads to a decrease in the attack's stealthiness, making it easily identified and filtered by mainstream federated defense algorithms, while also incurring higher attack costs (Xu et al., 2022; Xi et al., 2021; Yang et al., 2024b).

Besides, **graph adversarial attacks** can also cause the model to produce incorrect predictions. However, this method also has its limitations: **Challenge 3)** Due to the discrete nature of graph structures and the non-convexity of the optimization objective, it often suffers from slow convergence,

unstable optimization, or even failure to converge, leading to significant computational overhead and performance uncertainty (Li et al., 2025).

To address the above challenges, we propose **FedShift**, a novel distributed adversarial attack framework that incorporates ideas from backdoor attacks into adversarial attacks, addressing the limitations that each faces when used separately. It is divided into two stages: **Stage 1) Gentle Data Poisoning**. Traditional backdoor attacks that inject triggers and forcibly modify labels essentially construct an additional distribution of poisoned data and compel the model to learn a shortcut—a direct sample-to-label mapping based on the trigger. However, such shortcuts are proven to be easily smoothed out by the normal signals from benign clients or identified by defense algorithms (Blanchard et al., 2017; Bagdasaryan et al., 2020), which leads to Challenge 1 and Challenge 2. In contrast, before federated training, we design an adaptive generator for each malicious client to produce a perturbation we call a "shifter", which subtly pushes the embedding of the poisoned graph toward the decision boundary of the target class without crossing it (i.e., not enough to be classified as the target class). This gentle distributional shift makes the behavior of malicious clients almost indistinguishable from that of benign clients. Therefore, the backdoor signal it produces can effectively resist signal smoothing and ensure stealthiness, thereby solving Challenges 1 and 2. **Stage 2) Adversarial Perturbation Finding**. Traditional adversarial attacks perform optimization from scratch after federated training is complete, which leads to slow and unstable convergence (Li et al., 2025), i.e., Challenge 3. In contrast, after federated training, our method leverages the information from the global model, which has already been implanted with a backdoor, and uses the shifter generator trained in Stage 1 as an optimization starting point. Consequently, the process of finding the adversarial perturbation becomes both stable and efficient, thereby solving Challenge 3. During the final attack, we aggregate the different adversarial perturbations generated by multiple malicious clients to form the final adversarial sample, achieving a "$1 + 1 > 2$" effect.

We extensively evaluate our FedShift on six large-scale graph datasets, and the attack results excellently address the three major challenges mentioned above: 1) Compared to existing methods, the backdoor signal from FedShift is 80.5% to 90.6% less smoothed by the federated aggregation process, demonstrating its superior attack effectiveness in large-scale, multi-client real-world scenarios. 2) FedShift maintains the highest attack effectiveness when confronted with various mainstream defense algorithms, showcasing its remarkable robustness and stealthiness. 3) FedShift requires over 90% fewer training epochs to achieve the same Attack Success Rate (ASR) compared to the baseline method, highlighting its exceptional efficiency and stability.

In summary, the contributions of this work are threefold:

- We propose **FedShift**, a novel distributed adversarial attack with stealthiness, effectiveness, and efficiency.
- We resolve the dilemmas of existing attack paradigms through a two-stage process utilizing a novel distributional shift strategy and an optimized initial state.
- We pioneer an "implant-find" attack paradigm. To our knowledge, this is the first study to leverage information from the entire federated learning process within a unified framework.

## 2 RELATED WORK

**Federated Graph Learning**. Federated Graph Learning (FedGL) (He et al., 2022; Xie et al., 2023; Liu et al., 2024) is a novel paradigm that merges Federated Learning (FL) (McMahan et al., 2017) with Graph Neural Networks (GNNs) (Scarselli et al., 2008) to enable collaborative training on private graph data. However, the distributed nature of FedGL, coupled with the inherent complexity of graph data, introduces significant security vulnerabilities to malicious attacks (Dai et al., 2023; Wu et al., 2024; Yang et al., 2024a).

**Attacks on Federated Graph Learning**. Early backdoor attacks on FedGL utilized random subgraphs as triggers (Xu et al., 2022), which later evolved to use adaptive generators for improved effectiveness (Yang et al., 2024b). To enhance stealthiness, other works employed adversarial attacks to find adversarial samples through post-hoc optimization after federated training is complete (Li et al., 2025). Nevertheless, existing paradigms are often limited by a trade-off between effectiveness, stealth, and convergence speed. In contrast, our proposed **FedShift** overcomes these limitations through a novel two-stage process that achieves a stealthy, robust, and efficient attack.

## 3 PRELIMINARY

### 3.1 FEDERATED GRAPH LEARNING (FEDGL)

We represent a graph as $G = (\mathcal{V}, \mathcal{E}, \mathbf{X})$, with nodes $\mathcal{V}$, edges $\mathcal{E}$, and a node feature matrix $\mathbf{X} \in \mathbf{R}^{|\mathcal{V}| \times d}$. We focus on graph classification, where a GNN model $f$ learns a mapping $f : G \rightarrow y$ to a label $y \in \mathcal{Y}$.

Federated Graph Learning (FedGL) is a framework where $N$ clients ($C = \{c_1, \ldots, c_N\}$), each holding a private dataset $\mathcal{D}_i$, collaboratively train a global GNN model parameterized by $\theta$. The objective is to minimize the weighted average of their local loss functions $\mathcal{L}_i(\theta)$:

$$\min_\theta \mathcal{L}(\theta) = \sum_{i=1}^{N} \frac{|\mathcal{D}_i|}{|\mathcal{D}|} \mathcal{L}_i(\theta). \tag{1}$$

This is achieved through an iterative process. In each epoch $t$, clients first download the current global model $\theta_t$. They then compute local updates by training on their private data, yielding local models $\theta_t^i$. Finally, the server aggregates these local models to produce the next global model, $\theta_{t+1}$, often using Federated Averaging (FedAvg): $\theta_{t+1} = \frac{1}{N} \sum_{i=1}^{N} \theta_t^i$. This repeats until convergence.

### 3.2 ATTACKS ON FEDGL

Backdoor attacks on FedGL are centered on data poisoning. In this approach, malicious clients poison local data by injecting trigger subgraphs which can be static (Xu et al., 2022) or adaptively generated for each local graph by a trigger generator (Xi et al., 2021; Yang et al., 2024b) and relabel the graphs to the target class $y_t$.

Adversarial attacks on FedGL, in contrast, avoid data poisoning. Instead, attackers use their own clean data to perform post-hoc optimization after the federated training process is complete, learning to construct adversarial samples that find and exploit the inherent vulnerabilities of the converged global model (Li et al., 2025).

### 3.3 THREAT MODEL

Our threat model considers an attacker controlling a subset of malicious clients $C_M \subset C$ to achieve a stealthy, effective, and efficient attack. We assume the following:

- **Attacker's knowledge:** Malicious clients only know their own local data $\mathcal{D}_i$ and the global model parameters $\theta_t$ received from the server during FedGL training.
- **Attacker's capability:** Malicious clients can inject and continuously optimize shifters within their local training data throughout the whole federated process.
- **Attacker's objective:** The attacker aims to generate adversarial samples that are misclassified by the global model as the target label, without compromising the model's classification accuracy on clean data.

## 4 METHOD

To address the dilemmas of existing attacks on FedGL, we propose a novel two-stage distributed adversarial attack framework, named **FedShift**, whose overall workflow is illustrated in Figure 1. The goal of Stage 1, **Gentle Data Poisoning**, is to train an adaptive shifter generator for each malicious client before federated training begins. This generator produces a gentle distributional shift to solve the backdoor signal smoothing problem and improve attack stealthiness. Furthermore, the shifter generator can be fine-tuned online during the subsequent federated training process according to the dynamic changes of the global model. After federated training is complete, Stage 2, **Adversarial Perturbation Finding**, utilizes the shifter generator trained in Stage 1 as an optimization starting point to efficiently and stably optimize the shifter as an effective adversarial perturbation. During the final attack, adversarial perturbations from multiple malicious clients are aggregated to generate the adversarial sample, thereby triggering an effective attack.

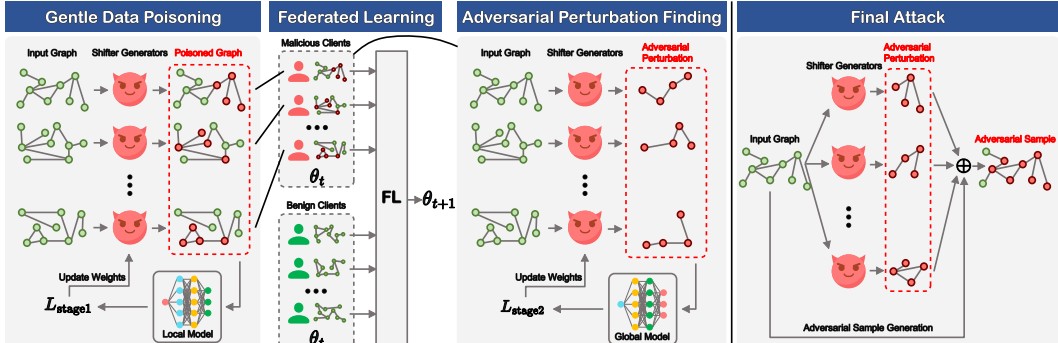

Figure 1: Pipeline of our two-stage adversarial attack. **Training**: In **Stage 1**, each malicious client uses local data to train a shifter generator and injects hidden shifters into the training data. During federated training, malicious clients inject the backdoor signal into the global model through the federated aggregation mechanism. In **Stage 2**, using the shifter generator trained in Stage 1 as a high-quality starting point, the shifter is further optimized as an effective adversarial perturbation leveraging the information from the global model. **Attack**: Aggregate adversarial perturbations from multiple malicious clients to generate the adversarial sample and trigger the attack.

## 4.1 STAGE 1: GENTLE DATA POISONING

The core of this stage is to train an adaptive shifter generator for each malicious client and inject hidden shifters into the training data before federated training. This involves two main steps. First, each malicious client trains a local GNN model using its local data to guide the subsequent training of the shifter generator. Second, each malicious client trains its local adaptive shifter generator to achieve the implantation of stealthy backdoor signals.

### 4.1.1 TRAINING THE LOCAL GNN MODEL

We adopt the Graph Attention Network (GAT) (Veličković et al., 2017) as the local GNN model. Through its attention mechanism, GAT offers both powerful modeling capabilities and excellent interpretability in graph representation learning. For each client $i$, its local GNN model $\theta_i^*$ is obtained by minimizing the loss on its local dataset $\mathcal{D}_i$:

$$\theta_i^* = \underset{\theta_i}{\arg\min} \, \mathcal{L}(\mathcal{D}_i; \theta_i). \tag{2}$$

This pre-trained local model $\theta_i^*$ will be used as a stable evaluation tool to extract high-quality graph embeddings and guide the selection of training graphs in the subsequent shifter generator training.

Notably, during the federated training process, the global GNN model received from the server can be optionally utilized to fine-tune the shifter generator. This enables it to leverage the rich information aggregated by the global model in each epoch, which is similar to the method in Opt-GDBA (Yang et al., 2024b).

### 4.1.2 ADAPTIVE SHIFTER GENERATOR

Our adaptive shifter generator comprises two core components: shifter position learning and shifter shape learning. The former identifies key nodes in the graph to maximize attack impact and enhance efficiency. The latter generates adaptive shifter to enable stealthy and effective backdoor implantation.

**1) Shifter position learning:** For a given input graph $G \in \mathcal{D}_i$, our goal is to select a subset of nodes $\mathcal{V}_p \subset \mathcal{V}$ to be injected with the shifter. Specifically, we introduce an efficient and low-complexity algorithm that calculates the clustering coefficient based on the type of graph data, its internal topological structure, and the weights of nodes and edges, to measure the influence of the nodes in the graph. The specific formulas is provided in the Appendix A.2. After calculating the

clustering coefficient values for all nodes in the graph, we select the set of nodes with the highest values as the subset of nodes $\mathcal{V}_p$ to be injected with the hidden shifter.

**2) Shifter Shape Learning:** After determining the node positions $\mathcal{V}_p$ for shifter injection, the objective of shape learning is to generate a specific shifter $\delta$ to achieve a stealthy attack, as outlined in Algorithm 1.

**i) Distributional proximity loss:** Unlike existing attack methods that directly modify labels to enforce target mappings, we propose a distributional proximity loss to ensure the attack's stealthiness. Specifically, for the $i$-th malicious client, we leverage its pre-trained local GNN model $\theta_i^*$ to extract high-dimensional graph embeddings. We define $\text{Enc}(\cdot; \theta_i^*)$ as a function that uses the model $\theta_i^*$ to extract the embedding vector from the output of the layer preceding the final classifier. We first obtain the embedding vector $v_p$ of a poisoned training graph $G_p$ (injected with a shifter), and the set of embedding vectors $\mathbf{V}_t$ for all target-class graphs in $\mathcal{D}_{y_t}$:

$$v_p = \text{Enc}(G_p; \theta_i^*),$$

$$\mathbf{V}_t = \{\text{Enc}(G_j; \theta_i^*) \mid G_j \in \mathcal{D}_{y_t}\}.$$

---

**Algorithm 1** Adaptive Shifter Generator Training

**Input**: Malicious client set $C_M$ with local dataset $\{\mathcal{D}_i\}_{i \in C_M}$ and pre-trained local GNN model $\{\theta_i^*\}_{i \in C_M}$, initial shifter generator parameters $\{\omega_i^0\}_{i \in C_M}$, target class $y_t$.
**Parameter**: Number of clusters $k$, training epochs $E$, learning rate $\eta$, number of nodes to perturb $n_{tri}$, loss weights $\lambda_{dist}, \lambda_{homo}, \lambda_{ce}$.
**Output**: Malicious clients' trained shifter generator parameters $\{\omega_i\}_{i \in C_M}$.

1: **for** each malicious client $i \in C_M$ **do**
2:     Select $\mathcal{D}_{y_t}, \mathcal{D}_{\text{train}} \subseteq \mathcal{D}_i$
3:     $\mathbf{V}_t \leftarrow \{\text{Enc}(G_j; \theta_i^*) \mid G_j \in \mathcal{D}_{y_t}\}$
4:     $\mathcal{C}_t \leftarrow \text{KMeans}(\mathbf{V}_t, k)$
5:     **for** each epoch $e = 1, \ldots, E$ **do**
6:         **for** each graph $G \in \mathcal{D}_{\text{train}}$ **do**
7:             $\delta \leftarrow G_{\text{gen}}^{(i)}(G, \mathcal{V}_p; \omega_i)$
8:             $G_p \leftarrow G \oplus \delta$
9:             $v_p \leftarrow \text{Enc}(G_p; \theta_i^*)$
10:           $c_{\text{near}} \leftarrow \underset{c_j \in \mathcal{C}_t}{\arg\min} \|v_p - c_j\|_2$
11:           $L_{dist} \leftarrow 1 - \cos(v_p, c_{\text{near}})$
12:           $L_{homo} \leftarrow \text{Homogeneity}(G_p, \mathcal{V}_p)$
13:           $L_{ce} \leftarrow \text{CrossEntropy}(f(G_p; \theta_i^*), \text{label}(G))$
14:           $L_{\text{stage1}} \leftarrow \lambda_{dist} L_{dist} + \lambda_{homo} L_{homo} + \lambda_{ce} L_{ce}$
15:           $\omega_i \leftarrow \omega_i - \eta \cdot \nabla_{\omega_i} L_{\text{stage1}}$
16:         **end for**
17:     **end for**
18: **end for**
19: **return** $\{\omega_i\}_{i \in C_M}$

---

Next, we apply the $k$-means clustering[1] to the set of target-class embeddings $\mathbf{V}_t$ to obtain $k$ cluster centroids $\mathcal{C}_t = \{c_1, \ldots, c_k\}$. For the poisoned graph's embedding vector $v_p$, we find the nearest cluster centroid $c_{\text{near}}$ in $\mathcal{C}_t$ with respect to the cosine distance. Finally, we define the distributional proximity loss $L_{dist}$ as the cosine distance between these two vectors, thereby minimizing the angular difference between them:

$$L_{dist} = 1 - \frac{v_p \cdot c_{\text{near}}}{\|v_p\|_2 \|c_{\text{near}}\|_2}. \tag{3}$$

By minimizing $L_{dist}$, we guide the generation of the shifter, causing the poisoned graph to gradually approach the distribution area of the target class in the feature space. Since there is no direct label modification or forced mapping construction, this distributional shift is gentle and stealthy.

It is worth noting that we only use training graphs that are correctly classified by the local GNN model $\theta_i^*$ from each malicious client and prioritize poisoning the training graphs that are farthest (with the largest cosine distance) from the target-class graphs in the feature space, in order to ensure an effective distributional shift.

**ii) Design of adaptive shifter generator:** As pointed out by Ding et al. (2025), solely modifying the features of poisoned nodes without altering the graph's edge connectivity can significantly enhance the attack stealthiness. Inspired by this insight, our adaptive shifter generator $G_{\text{gen}}$'s objective is, given an original graph $G_i = (\mathcal{V}_i, \mathcal{E}_i, \mathbf{X}_i)$ and a determined set of poisoned node positions $\mathcal{V}_p$, to generate the optimal feature perturbation $\Delta \mathbf{X}_p$ for these nodes.

Regarding the specific network architecture of $G_{\text{gen}}$, we found that simpler models such as Multi-Layer Perceptrons (MLPs) and Graph Convolutional Networks (GCNs) (Kipf & Welling, 2016) are

---

[1]We employ the K-means algorithm for its well-established efficiency and effectiveness. Although more advanced techniques exist, our selection was guided not by the need for a superior clustering algorithm in general, but by the specific requirements of our task: learning the local trigger shape.

sufficient to effectively capture the local dependencies required for generating perturbations. These models have lower computational overhead and are more suitable for efficiency-sensitive scenarios like federated learning. $G_{\text{gen}}$ can be formally represented as:

$$\Delta \mathbf{X}_p = G_{\text{gen}}(G_i, \mathcal{V}_p). \tag{4}$$

We define the attack pattern, which consists of the positions $\mathcal{V}_p$ and the corresponding feature perturbations $\Delta \mathbf{X}_p$, as the shifter $\delta = (\mathcal{V}_p, \Delta \mathbf{X}_p)$. Ultimately, the poisoned graph $G_p$ is generated by applying this shifter to the original graph $G_i$, denoted as $G_p = G_i \oplus \delta$.

To further enhance stealthiness, we introduce two supplementary loss terms, Homogeneity Loss and Boundary-Balancing Cross-Entropy Loss. The **Homogeneity Loss ($L_{\text{homo}}$)** is based on the graph homophily assumption, which posits that connected nodes should have similar features:

$$L_{\text{homo}} = \frac{1}{|\mathcal{E}|} \sum_{(u,v) \in \mathcal{E}} \max(0, \tau - \text{sim}(\mathbf{x}_u, \mathbf{x}_v)), \tag{5}$$

where $|\mathcal{E}|$ is the number of edges in graph, $\text{sim}(\cdot, \cdot)$ is the cosine similarity function, and $\tau$ is a predefined similarity threshold.

We also design the **Boundary-Balancing Cross-Entropy Loss ($L_{\text{ce}}$)** as a key balancing term, which can prevent the distributional shift from easily crossing the decision boundary. It calculates the cross-entropy loss of the shifter-injected graph $G_p$ being predicted as its original correct label $y_s$ by the local model $\theta_i^*$:

$$L_{\text{ce}} = \text{CrossEntropy}(f(G_p; \theta_i^*), y_s). \tag{6}$$

Ultimately, the optimization objective for our adaptive shifter generator model is to minimize the following loss:

$$L_{\text{stage1}} = \lambda_{\text{dist}} L_{\text{dist}} + \lambda_{\text{homo}} L_{\text{homo}} + \lambda_{\text{ce}} L_{\text{ce}}, \tag{7}$$

where $\lambda_{\text{dist}}$, $\lambda_{\text{homo}}$, and $\lambda_{\text{ce}}$ are coefficients to balance the different objectives.

### 4.2 STAGE 2: ADVERSARIAL PERTURBATION FINDING

After all clients have completed the FedGL training process and a converged global model $\theta^*$ is obtained, the attack enters the final adversarial perturbation finding stage. Unlike NI-GDBA (Li et al., 2025) which optimizes from scratch, we utilize the shifter generator trained in Stage 1 as a high-quality starting point and leverage the rich information aggregated in the global model to efficiently and stably fine-tune the shifter as the final effective adversarial perturbation.

The specific optimization process is as follows: the attacker freezes the parameters of the final global model $\theta^*$ and continues to train (or fine-tune) the trigger generator. The optimization objective shifts to maximizing the attack success rate, to ensure that during the final attack, the embedding distribution of the adversarial example—formed by aggregating adversarial perturbations from multiple malicious clients—can cross the decision boundary in the feature space, thereby achieving an effective attack. The optimization is guided by a loss function composed of a standard cross-entropy attack loss $L_{\text{attack}}$ and the homogeneity loss $L_{\text{homo}}$:

$$L_{\text{attack}} = \text{CrossEntropy}(f(G_p; \theta^*), y_t), \tag{8}$$

where $G_p$ is a graph injected with the shifter, $y_t$ is the attacker's target label. Ultimately, the optimization objective for our adversarial perturbation finding stage is to minimize the following loss:

$$L_{\text{stage2}} = L_{\text{attack}} + \lambda_{\text{homo}} L_{\text{homo}}, \tag{9}$$

During the final attack, adversarial perturbations from multiple malicious clients are aggregated to generate the adversarial sample, thereby triggering an effective attack.

## 5 EXPERIMENT

Next, we conduct an empirical study of our FedShift to answer the following key questions:

- **Q1:** Can FedShift resist the signal smoothing effect inherent to the FedGL training process?
- **Q2:** Facing existing FedGL defense algorithms, can FedShift ensure attack stealthiness?
- **Q3:** Can FedShift achieve both rapid and stable convergence during the perturbation finding stage?

## 5.1 EXPERIMENTAL SETUP

We implement FedShift on FedGL using the PyTorch framework. All experiments are conducted on a server equipped with 8 NVIDIA 4090 GPUs. Each experiment is repeated five times with different random seeds to obtain averaged attack results.

**Datasets and training/testing sets:** Compared to existing work, we have expanded the scale of our datasets to better simulate real-world scenarios with six large-scale graph datasets that cover four common real-world domains: small compound molecules (Morris et al., 2020), bioinformatics (Morris et al., 2020), social networks (Dou et al., 2021), and finance (Zhou et al., 2022). Detailed information can be find in Appendix A.3. For each dataset, we randomly sample 80% of the data instances as the training dataset and the rest as the testing dataset.

**Attack baselines:** We compare FedShift with current state-of-the-art backdoor attack methods, including Rand-GDBA (Xu et al., 2022), GTA (Xi et al., 2021), and Opt-GDBA (Yang et al., 2024b), as well as the adversarial attack method NI-GDBA (Li et al., 2025).

**Parameter settings:** In all experiments, we default to set the trigger node ratio to $n_{tri} = 0.1$ for all attack methods. For our FedShift, we set the number of clusters to $k = 3$. The Graph Attention Network (GAT) (Veličković et al., 2017) is used as the backbone classifier for all experiments, with a total of 40 federated training epochs. For each Q, the specific attack settings are as follows:

- **Q1 settings:** To evaluate each attack's resilience to federated smoothing from benign clients, we adopt a low attack budget, i.e., poisoned graph ratio $p = 0.1$ and poisoned node feature dimension ratio $f = 0.1$. We fix the number of malicious clients at $|C_M| = 4$ (for the DD dataset, which has fewer graphs, we set $|C_M| = 2$) and increase the number of benign clients to set the malicious clients proportion ($|C_M|/N$) as 0.2, 0.1, and 0.05.

- **Q2 settings:** To test the evasion capability against FedGL defense algorithms, we fix the attack budget at a moderate intensity, i.e., $p = 0.2$, $f = 0.2$ and a total of $N = 40$ clients with $|C_M| = 4$ malicious clients ($N = 20, |C_M| = 2$ for the DD dataset). We test against three mainstream federated defense algorithms: foolsgold (Fung et al., 2020), fedkrum (Blanchard et al., 2017), and fedbulyan (Guerraoui et al., 2018).

- **Q3 settings:** To verify the attack efficiency under stringent conditions, we adopt a low attack budget of $p = 0.1$, $f = 0.1$. We set $N = 40$ and $|C_M| = 4$ ($N = 20, |C_M| = 2$ for the DD dataset). We compare the convergence of both our full FedShift model and an ablation variant without federated online fine-tuning against the adversarial attack method NI-GDBA.

**Evaluation metrics:** We use the **Attack Success Rate (ASR)** to evaluate the attack's effectiveness and the **Original Task Accuracy (OA)** to evaluate the GNN model's performance. We also propose the **Adaptive Attack Score (AAS)** to evaluate the attack's comprehensive effectiveness:

$$\text{AAS} = \text{ASR} \cdot \text{OA}^{\text{ASR}}. \tag{10}$$

This metric is designed to prioritize a high ASR, as an attack with only high OA is meaningless. Therefore, the model's OA contributes significantly to the AAS score only when ASR is sufficiently high. Otherwise, the score is determined almost entirely by the ASR itself.

## 5.2 EXPERIMENTAL RESULTS

### 5.2.1 MAIN RESULTS OF THE COMPARED ATTACKS:

The main experimental results are presented in Tables 1, 2 and Figures 2, 3. For more results, please refer to the Appendix A.4.1. We derive the following key observations:

1. **For Q1: FedShift effectively resists the signal smoothing inherent in the federated learning process.** As shown in Table 1 and Figure 2, as the proportion of malicious clients decreases from 0.2 to 0.05, the ASR of traditional methods drops by an average of **25.6% to 53.3%** due to signal smoothing. In contrast, our model's ASR drops by **less than 5%** and consistently maintains the best attack effectiveness. This demonstrates our method's strong resistance to the backdoor signal smoothing, strongly answering Q1.

Table 1: Attack results in the Q1 setting with the malicious client proportions $|C_M|/N = 0.2$.

| Methods | Rand-GDBA | | | GTA | | | Opt-GDBA | | | NI-GDBA | | | FedShift (Ours) | | |
|---|---|---|---|---|---|---|---|---|---|---|---|---|---|---|---|
| Metrics | AAS | ASR | OA | AAS | ASR | OA | AAS | ASR | OA | AAS | ASR | OA | AAS | ASR | OA |
| DD | 0.01 | 0.01 | 0.63 | 0.06 | 0.06 | 0.64 | 0.03 | 0.03 | **0.65** | 0.44 | 0.59 | 0.61 | **0.58** | **0.88** | 0.62 |
| NCI109 | 0.39 | 0.48 | 0.63 | 0.62 | **0.98** | 0.62 | 0.56 | 0.84 | 0.61 | 0.50 | 0.66 | 0.64 | **0.65** | **0.98** | **0.66** |
| Mutagenicity | 0.34 | 0.38 | 0.73 | 0.71 | **0.99** | 0.71 | 0.46 | 0.56 | 0.72 | 0.11 | 0.12 | 0.73 | **0.74** | **0.99** | **0.75** |
| FRANKENSTEIN | 0.26 | 0.31 | 0.58 | 0.58 | **1.00** | 0.58 | 0.54 | 0.85 | 0.59 | **0.61** | **1.00** | **0.61** | **0.61** | **1.00** | **0.61** |
| Eth-Phish&Hack | 0.05 | 0.05 | 0.91 | 0.91 | 0.99 | 0.92 | 0.16 | 0.16 | 0.91 | 0.92 | **1.00** | 0.92 | **0.93** | **1.00** | **0.93** |
| Gossipcop | 0.58 | 0.79 | 0.68 | 0.68 | **1.00** | 0.68 | 0.67 | 0.94 | 0.70 | 0.68 | **1.00** | 0.68 | **0.76** | 0.99 | **0.76** |

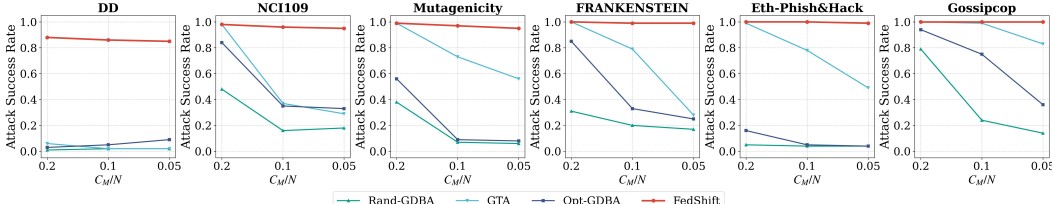

Figure 2: Attack results in the Q1 setting under defferent malicious client proportions $|C_M|/N$.

Table 2: Attack results in the Q2 setting under different defenses.

| Methods | Rand-GDBA | | | GTA | | | Opt-GDBA | | | NI-GDBA | | | FedShift (Ours) | | |
|---|---|---|---|---|---|---|---|---|---|---|---|---|---|---|---|
| Defenses | Foo. | Kru. | Bul. | Foo. | Kru. | Bul. | Foo. | Kru. | Bul. | Foo. | Kru. | Bul. | Foo. | Kru. | Bul. |
| DD | 0.02 | 0.30 | 0.30 | 0.11 | 0.30 | 0.30 | 0.06 | 0.18 | 0.18 | **0.62** | 0.44 | 0.44 | **0.62** | **0.45** | **0.45** |
| NCI109 | 0.14 | 0.49 | 0.49 | 0.44 | 0.45 | 0.45 | 0.39 | 0.47 | 0.47 | 0.57 | **0.50** | **0.50** | **0.59** | **0.50** | **0.50** |
| Mutagenicity | 0.07 | 0.19 | 0.17 | 0.48 | 0.48 | 0.48 | 0.54 | 0.48 | 0.48 | 0.40 | 0.46 | 0.46 | **0.58** | **0.49** | **0.49** |
| FRANKENSTEIN | 0.17 | 0.32 | 0.31 | **0.61** | 0.29 | 0.29 | 0.40 | 0.37 | 0.37 | **0.61** | **0.54** | **0.54** | **0.61** | **0.54** | **0.54** |
| Eth-Phish&Hack | 0.04 | 0.42 | 0.42 | 0.82 | 0.75 | 0.75 | 0.06 | 0.44 | 0.44 | 0.78 | **0.78** | **0.78** | **0.83** | **0.78** | **0.78** |
| Gossipcop | 0.15 | 0.00 | 0.00 | 0.80 | 0.00 | 0.02 | 0.77 | 0.00 | 0.00 | **0.83** | 0.50 | 0.50 | **0.83** | **0.54** | **0.54** |

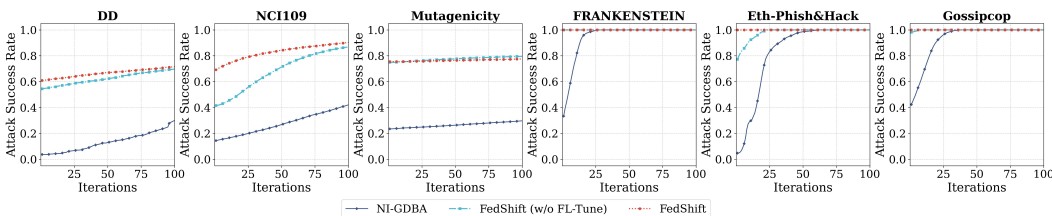

Figure 3: Attack results in the adversarial perturbation finding stage in the Q3 setting.

2. **For Q2: FedShift exhibits high stealthiness against existing federated learning defense algorithms.** As shown in Table 2, after introducing defense algorithms, our model consistently maintains the best attack effectiveness with the highest AAS, outperforming the strongest baseline method in each scenario by an average of **4.9%**. This demonstrates that its stealthiness can effectively evade existing defense algorithms, thereby answering Q2.

3. **For Q3: FedShift converges more efficiently to a better result.** As shown in Figure 3, compared to NI-GDBA, which optimizes from scratch, our model without federated tuning and our full model require **90.3%** and **98.3%** fewer epochs, respectively, to achieve the same ASR with an optimized starting point. The superior convergence speed of the full model demonstrates that the effectiveness of federated optimization. The whole result verifies the efficiency of our framework, providing a satisfactory answer to Q3.

### 5.2.2 IMPACT OF HYPERPARAMETERS ON OUR FEDSHIFT

In this set of experiments, we will study in-depth the impact of the important hyperparameters on our FedShift. For more results, please refer to the Appendix A.4.2.

**Impact of the poisoned node feature dimension ratio $f$ and the poisoned graph ratio $p$:** As shown in Figure 4, as $f$ increases from 0.1 to 0.3 with $p = 0.2$, the AAS and ASR of our

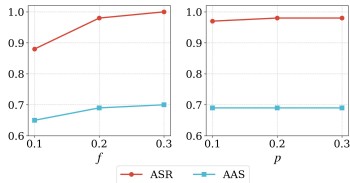

Figure 4: Impact of $f$ and $p$ of our FedShift.

method steadily improve, indicating that FedShift can effectively utilize more poisoned node features to achieve better attack effectiveness. Meanwhile, as $p$ increases from 0.1 to 0.3 with $f = 0.2$, the AAS and ASR remain at a high level, demonstrating that FedShift is low in dependence on the number of poisoned samples and can be effective with only a small amount of poisoned graphs.

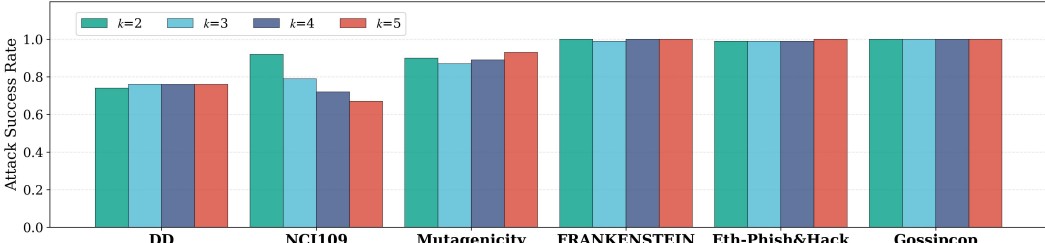

Figure 5: Attack results of FedShift with the number of target-class clusters $k$ ranging from 2 to 5.

**Impact of the number of clusters $k$:** As shown in Figure 5, the sensitivity of our attack to the number of clusters $k$ varies between datasets. For datasets such as NCI109 and Mutagenicity, the ASR fluctuates with $k$, suggesting that their target-class embeddings may have a complex or multi-modal structure where the choice of $k$ crucial. In contrast, for datasets like Eth-Phish&Hack and Gossipcop, the attack effectiveness is almost unaffected by $k$, demonstrating our method's robustness on datasets where the target-class embeddings likely form a single, dense cluster.

### 5.2.3 ABLATION STUDY

Table 3: Performance comparison of different components of our FedShift.

| Methods | Stage 1 Only | | | Stage 1 + FL-Tune | | | Stage 1 + Stage 2 | | | Full FedShift | | |
|---|---|---|---|---|---|---|---|---|---|---|---|---|
| Metrics | AAS | ASR | OA | AAS | ASR | OA | AAS | ASR | OA | AAS | ASR | OA |
| DD | 0.34 | 0.41 | **0.62** | 0.38 | 0.48 | **0.62** | 0.52 | 0.73 | **0.62** | **0.53** | **0.76** | **0.62** |
| NCI109 | 0.33 | 0.42 | 0.56 | 0.38 | 0.51 | **0.57** | **0.49** | **0.77** | 0.56 | 0.47 | 0.69 | **0.57** |
| Mutagenicity | 0.49 | 0.63 | **0.66** | 0.49 | 0.64 | **0.66** | 0.52 | 0.69 | **0.66** | 0.51 | 0.67 | **0.66** |
| FRANKENSTEIN | 0.40 | 0.51 | **0.61** | 0.58 | 0.91 | **0.61** | 0.61 | 0.99 | **0.61** | 0.61 | 0.99 | **0.61** |
| Eth-Phish&Hack | 0.66 | 0.72 | **0.89** | 0.86 | 0.97 | 0.89 | 0.88 | 0.99 | 0.89 | 0.88 | 0.99 | 0.89 |
| Gossipcop | **0.83** | 0.98 | **0.84** | **0.83** | 0.99 | **0.84** | **0.83** | 0.99 | **0.84** | 0.83 | **1.00** | **0.84** |

In this experiment, we examine the necessity of each component in our two-stage framework of FedShift under stringent conditions with a low attack budget of $p = 0.1, f = 0.1$. The results are shown in Table 3. We compare four settings: using only the first stage (Stage 1 Only), adding federated online fine-tuning (Stage 1 + FL-Tune) or adding the second stage (Stage 1 + Stage 2), and the complete two-stage method (Full FedShift). The results show that compared to Stage 1 Only, adding FL-Tune and Stage 2 increases the average AAS by 17.0% and 32.2% respectively, demonstrating the significant impact of both components, particularly Stage 2. More strikingly, the Stage 1 + Stage 2 configuration achieves a nearly identical level of effectiveness to the full model, which in turn improves upon the Stage 1 + FL-Tune setup by 12.5%. This suggests that incorporating Stage 2 is the key and sufficient step for maximizing the attack's effectiveness.

## 6 CONCLUSION

From an attacker's perspective, we examine the security of FedGL and propose a novel, two-stage distributed adversarial attack method that is effective, stealthy, and efficient. Our method implants hidden and learnable shifters into the training graph via a distributional shift in the first stage, and subsequently uses them as a starting point in the second stage to efficiently find adversarial perturbations. During the final attack, these perturbations are aggregated from multiple clients to form the final effective adversarial sample. Our experiments demonstrate that this method can achieve high attack effectiveness, effectively evade mainstream defense algorithms, and significantly improve attack efficiency. This work provides a new perspective for future security and defense research on FedGL.

## 7 ETHICS STATEMENT

The FedShift framework we propose reveals a novel and highly stealthy security vulnerability in Federated Graph Learning systems. The fundamental purpose of this research is to raise security awareness in the field, not to facilitate malicious activities. We firmly believe that a deep understanding of advanced attack methods is a necessary prerequisite for constructing stronger and more robust defense strategies. In line with the principles of responsible academic disclosure, we publicize our findings to inspire and aid the research community in developing defense mechanisms capable of effectively countering such two-stage attacks. We encourage our peers in the academic community to leverage the insights from this research to collectively advance Federated Graph Learning towards a more secure and trustworthy future.

## 8 REPRODUCIBILITY

To ensure the full reproducibility of our research findings, we provide comprehensive supporting materials. The complete source code for all experiments, including model implementations and the scripts used to generate the results, will be made publicly available in a repository upon the paper's publication. Detailed information regarding the experimental setup, including all hyperparameter configurations and model architectures, is described in the main text. All datasets used in this study are public benchmarks, and their statistics and descriptions are also provided in the Appendix. We hope these resources will facilitate the verification and replication of our work by the research community and encourage further exploration in this area.

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

## A Appendix

### A.1 The Use of Large Language Models

During the writing process of this paper, we used a Large Language Model (LLM) as a writing assistance tool. Its use was limited to language polishing, style optimization, and grammar checks. We explicitly state that all core research ideas, theoretical derivations, experimental design, and result analysis were conducted independently by the human authors. We take full responsibility for the entire content of the paper and have carefully reviewed and verified the accuracy and originality of all of its statements.

### A.2 Clustering Coefficient Calculation

In our work, the influence of nodes within a graph is measured by the clustering coefficient. The formulas are detailed below:

- **For unweighted graphs:** The clustering coefficient $c(u)$ of a node $u$ is the fraction of possible triangles through the node that exist. It is calculated as:

$$c(u) = \frac{2 \cdot \mathcal{T}(u)}{d(u)(d(u) - 1)}, \tag{11}$$

where $\mathcal{T}(u)$ denotes the number of triangles through node $u$, and $d(u)$ is the degree of node $u$. When $d(u) < 2$, the value of $c(u)$ is set to 0.

- **For weighted graphs:** The clustering coefficient is defined as the geometric average of the subgraph edge weights:

$$c(u) = \frac{1}{d(u)(d(u) - 1)} \sum_{v,w} (\hat{\omega}_{uv} \hat{\omega}_{uw} \hat{\omega}_{vw})^{1/3}, \tag{12}$$

where $\hat{\omega}_{uv}$ is the edge weight $\omega_{uv}$ normalized by the maximum weight in the network.

- **For directed graphs:** The clustering coefficient considers the directionality of edges as:

$$c(u) = \frac{2 \cdot \mathcal{T}_d(u)}{d^{\text{tot}}(u)(d^{\text{tot}}(u) - 1) - 2d^{\leftrightarrow}(u)}, \tag{13}$$

where $\mathcal{T}_d(u)$ is the number of directed triangles through node $u$, $d^{\text{tot}}(u)$ is the sum of the in-degree and out-degree of $u$, and $d^{\leftrightarrow}(u)$ is the reciprocal degree of $u$.

### A.3 Dataset Statistics and Descriptions

Table 4: Statistics of the benchmark datasets used in our experiments.

| Datasets | Graphs | Classes | Class Ratio | Avg. Nodes | Avg. Edges | Node Feats. |
|---|---|---|---|---|---|---|
| DD | 1,178 | 2 | 691 / 487 | 284.3 | 715.7 | 89 |
| NCI109 | 4,127 | 2 | 2,048 / 2,079 | 29.7 | 32.1 | 38 |
| Mutagenicity | 4,337 | 2 | 2,401 / 1,936 | 30.3 | 30.8 | 14 |
| FRANKENSTEIN | 4,337 | 2 | 1,936 / 2,401 | 16.9 | 17.9 | 780 |
| Eth-Phish&Hack | 5,070 | 2 | 2,535 / 2,535 | 37.8 | 111.3 | 5,000 |
| Gossipcop | 5,464 | 2 | 2,732 / 2,732 | 57.5 | 56.5 | 310 |

Table 4 shows the detailed statistics of our six large-scale graphs datasets. The descriptions of these datasets are as below:

**DD:** It is a dataset of protein structures, where each protein is represented as a graph. The task is to classify each structure as either an enzyme or a non-enzyme.

**NCI109:** It consists of chemical compounds screened for activity against ovarian cancer cell lines. Each graph represents a compound, and the task is to classify whether it is active in an anti-cancer screen.

**Mutagenicity:** It is a toxicology dataset where each graph represents a molecular structure. The task is to classify each molecule as mutagenic or non-mutagenic.

**FRANKENSTEIN:** It consists of molecular graphs with a binary label indicating their toxicological properties. Each vertex is labeled by the chemical atom symbol and edges by the bond type.

**Eth-Phish&Hack:** It derives from Ethereum transactions. Each graph represents an account's transaction subgraph, and the task is to identify accounts associated with phishing or hacking activities. Due to GPU memory constraints, we only select 5,000 node feature dimensions for our experiments.

**Gossipcop:** This dataset is used for the detection of fake news in the social networks domain. Each graph represents the propagation network of a news story on Twitter, and the task is to classify the news as either real or fake.

### A.4 ADDITIONAL EXPERIMENTAL RESULTS

#### A.4.1 MORE ATTACK RESULTS

Table 5: Attack results in the Q1 setting with the malicious client proportions $|C_M|/N = 0.1$.

| Methods | Rand-GDBA | | | GTA | | | Opt-GDBA | | | NI-GDBA | | | FedShift (Ours) | | |
| Metrics | AAS | ASR | OA | AAS | ASR | OA | AAS | ASR | OA | AAS | ASR | OA | AAS | ASR | OA |
|---|---|---|---|---|---|---|---|---|---|---|---|---|---|---|---|
| DD | 0.02 | 0.02 | 0.63 | 0.02 | 0.02 | 0.64 | 0.05 | 0.05 | **0.66** | 0.48 | 0.65 | 0.63 | **0.57** | **0.86** | 0.62 |
| NCI109 | 0.15 | 0.16 | 0.60 | 0.31 | 0.37 | 0.61 | 0.30 | 0.35 | 0.60 | 0.44 | 0.59 | 0.60 | **0.64** | **0.96** | **0.66** |
| Mutagenicity | 0.07 | 0.07 | 0.65 | 0.53 | 0.73 | 0.65 | 0.09 | 0.09 | 0.65 | 0.26 | 0.29 | 0.65 | **0.72** | **0.97** | **0.74** |
| FRANKENSTEIN | 0.18 | 0.20 | **0.61** | 0.53 | 0.79 | **0.61** | 0.28 | 0.33 | **0.61** | 0.61 | **1.00** | **0.61** | 0.61 | 0.99 | **0.61** |
| Eth-Phish&Hack | 0.04 | 0.04 | 0.87 | 0.70 | 0.78 | 0.87 | 0.05 | 0.05 | 0.87 | 0.87 | **1.00** | 0.87 | **0.88** | **1.00** | **0.89** |
| Gossipcop | 0.23 | 0.24 | 0.83 | 0.81 | 0.99 | 0.82 | 0.65 | 0.75 | 0.83 | 0.82 | **1.00** | 0.82 | **0.83** | **1.00** | **0.84** |

Table 6: Attack results in the Q1 setting with the malicious client proportions $|C_M|/N = 0.05$.

| Methods | Rand-GDBA | | | GTA | | | Opt-GDBA | | | NI-GDBA | | | FedShift (Ours) | | |
| Metrics | AAS | ASR | OA | AAS | ASR | OA | AAS | ASR | OA | AAS | ASR | OA | AAS | ASR | OA |
|---|---|---|---|---|---|---|---|---|---|---|---|---|---|---|---|
| DD | 0.02 | 0.02 | 0.64 | 0.02 | 0.02 | 0.65 | 0.08 | 0.09 | **0.68** | 0.52 | 0.71 | 0.65 | **0.57** | **0.85** | 0.62 |
| NCI109 | 0.16 | 0.18 | 0.59 | 0.25 | 0.29 | 0.60 | 0.28 | 0.33 | 0.59 | 0.46 | 0.65 | 0.59 | **0.64** | **0.95** | **0.66** |
| Mutagenicity | 0.06 | 0.06 | 0.63 | 0.43 | 0.56 | 0.63 | 0.07 | 0.08 | 0.63 | **0.71** | **0.95** | **0.74** | **0.71** | **0.95** | **0.74** |
| FRANKENSTEIN | 0.16 | 0.17 | **0.60** | 0.24 | 0.28 | **0.60** | 0.22 | 0.25 | **0.60** | **0.60** | **1.00** | **0.60** | 0.59 | 0.99 | **0.60** |
| Eth-Phish&Hack | 0.04 | 0.04 | 0.86 | 0.45 | 0.49 | 0.86 | 0.04 | 0.04 | 0.86 | **0.86** | 0.99 | **0.87** | **0.86** | 0.99 | **0.87** |
| Gossipcop | 0.14 | 0.14 | **0.81** | 0.69 | 0.83 | 0.80 | 0.33 | 0.36 | 0.80 | 0.80 | **1.00** | 0.80 | **0.81** | **1.00** | **0.81** |

Tables 5 and 6 present the attack results for the Q1 setting with lower malicious client proportions ($|C_M|/N = 0.1$ and $|C_M|/N = 0.05$). The results show that when the proportion of malicious clients is low, the ASR and AAS values of the baseline methods drop significantly. In contrast, our method maintains high ASR and AAS, demonstrating its strong resistance to backdoor signal smoothing and further answering Q1.

#### A.4.2 MORE SENSITIVITY ANALYSES FOR KEY HYPERPARAMETERS

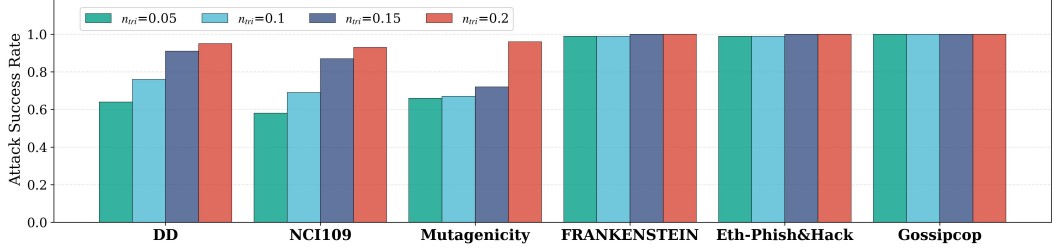

Figure 6: Attack results of FedShift with the trigger node ratio $n_{tri}$ ranging from 0.05 to 0.20.

**Impact of the Trigger Node Ratio $n_{tri}$:** As shown in Figure 6, the impact of the trigger node ratio $n_{tri}$ varies across different datasets. For datasets DD, NCI109 and Mutagenicity, the ASR shows a gradual increase as $n_{tri}$ is raised. This indicates that for these graphs data, our model can effectively leverage a larger set of perturbed nodes to progressively enhance the attack's effectiveness.

Conversely, for datasets FRANKENSTEIN, Eth-Phish&Hack and Gossipcop, the ASR remains consistently high and is largely unaffected by an increase in $n_{tri}$. This suggests that on these datasets, our method is highly efficient, capable of achieving a near-optimal attack effect by modifying only a very limited number of nodes.

