# OpenReview forum: "Hide and Find: A Distributed Adversarial Attack on Federated Graph Learning"
_ICLR.cc/2026/Conference — ICLR 2026 Conference Withdrawn Submission_

### Official Review · Reviewer_GoYE · 2025-10-27

**Soundness:** 3
**Presentation:** 2
**Contribution:** 2
**Rating:** 4
**Confidence:** 4

**Summary:**

This paper introduces FedShift, a two-stage distributed adversarial attack on Federated Graph Learning (FedGL). The attack framework aims to overcome key limitations of existing attack paradigms such as signal smoothing during aggregation, high computational cost, and detectability by integrating concepts from both backdoor and adversarial attacks. The proposed “Hide and Find” mechanism includes two stages in which the first stage introduces a learnable shifter generator that injects hidden, low-intensity perturbations into training data to cause subtle feature-space drifts toward a target class boundary. The second stage leverages the trained shifter and global model information post-training to efficiently fine-tune the perturbation for final attack execution. Extensive experiments on six large-scale datasets demonstrate that FedShift improves attack efficiency compared to other methods.

**Strengths:**

- The paper addresses a critical and emerging security issue in federated graph learning, a field gaining significant attention in federated and distributed AI research. The problem is well-motivated and clearly articulated.

- The formulation, notably Algorithm 1 for adaptive shifter generator training, and loss design are mathematically reasoned.

- The significance of this work is notable. FedShift reveals a new and realistic attack surface in distributed graph learning, which can have important implications for designing future defenses in federated systems. By proposing an attack that is both effective and stealthy, the paper advances the understanding of adversarial vulnerabilities in a rapidly growing field.

**Weaknesses:**

- The paper primarily provides an empirical demonstration of the proposed attack’s effectiveness but lacks theoretical grounding. For example, there is no formal analysis of complexity and why the two-stage design ensures convergence or how the “gentle shift” quantitatively balances stealth and effectiveness. Providing theoretical intuition, even in simplified settings (e.g., convergence bounds or an analysis of the distributional shift dynamics), would greatly strengthen the paper’s technical depth theoretically.

- Although the experiments cover six datasets, all are standard benchmarks under IID client data splits. Federated environments are typically non-IID, with varying client graph distributions and participation rates. Testing FedShift in such settings (e.g., heterogeneous feature or label distributions) would better demonstrate its robustness and practical viability.

- The related work section is relatively brief and does not sufficiently discuss existing defense mechanisms in Federated Graph Learning (FedGL). A more comprehensive review of defenses would help clarify the broader research landscape and better articulate the motivation for introducing this new type of attack. In its current form, the section focuses primarily on prior attack methods but lacks a balanced discussion that positions FedShift in relation to both attack and defense strategies. Additionally, including a comparative summary table (e.g., in the appendix) outlining the main characteristics, assumptions, and differentiating factors between FedShift and prior attacks would make the paper’s contributions and novelty clearer and more compelling.

- The paper would benefit from a brief limitations section outlining the current method’s drawbacks, such as scalability or attacker assumptions, and suggesting future extensions. Including remarks on potential defense strategies would also provide a more balanced and forward-looking perspective.

- The defense evaluation is limited to three well-known algorithms. Incorporating more recent adaptive or certified defense frameworks (e.g., anomaly-based or graph-structure defense) would enhance the comprehensiveness of the results.

- Although the authors highlight a 90% reduction in optimization time, it remains unclear how costly Stage 1 shifter training is in large-scale or resource-limited environments. Reporting per-client overhead or wall-clock time would provide practical insights.

- [Minor] Some figures (Figs. 2-4) could be visually clearer with larger fonts and consistent legends for better clarity and understanding

**Questions:**

1. All experiments appear to assume IID data partitions. How would the method behave under non-IID distributions. For example, heterogeneous node features or label skews among clients? Have the authors explored or plan to explore this case to verify robustness in more practical scenarios?

2. Could the authors comment on how FedShift might perform against newer structure-aware or representation-based defenses that analyze graph topology anomalies rather than gradient statistics? Would the “hidden shifter” still remain undetected?

3. How sensitive is the proposed FedShift to the lower number of malicious clients? Does FedShift maintain stealth and success under extremely small malicious proportions (for example <5%)?

4. Is the convex combination of loss terms (L_dist, L_homo, L_ce in Eq.7) tuned per dataset or fixed globally? Some discussion on hyperparameter sensitivity is required to understand the effectiveness of FedShift.

5. **Questions on Threat Model**
- Could the authors clarify whether the threat model is **white-box, grey-box, or black-box**?
- Since attackers can access global model parameters ($\theta_t$), why is a **two-stage attack** needed instead of directly poisoning local updates or model parameters?
- Are attackers restricted from modifying gradients or model updates, if so, why?
- Please provide references or justification showing that these assumptions reflect **practical FedGL threat model scenarios**.

---

### Official Review · Reviewer_UVgg · 2025-10-30

**Soundness:** 2
**Presentation:** 2
**Contribution:** 2
**Rating:** 4
**Confidence:** 3

**Summary:**

This paper proposes FedShift, a novel two-stage distributed adversarial attack for Federated Graph Learning (FedGL). The key idea is to first implant a “shifter” during pre-training to subtly push graph representations toward the target decision boundary (“Hide” stage), and then refine these perturbations post-training (“Find” stage) to generate effective adversarial samples. The authors claim that this two-phase design improves attack stealthiness, resistance to aggregation smoothing, and efficiency, achieving up to 90% reduction in time cost and strong evasion of several defense mechanisms (Foolsgold, Krum, Bulyan). Experiments on six graph datasets (DD, NCI109, Mutagenicity, FRANKENSTEIN, Eth-Phish&Hack, Gossipcop) show high Attack Success Rate (ASR) and stability across settings.

**Strengths:**

+ Comprehensive Experiments: The paper evaluates on six datasets across multiple domains, with ablations and sensitivity analyses that show consistent performance gains.
+ Strong Empirical Results: FedShift achieves notably higher attack success and efficiency compared to baselines, even under standard defense mechanisms.

**Weaknesses:**

- Limited Novelty: The approach mainly combines known techniques (backdoor + adversarial optimization) without substantial theoretical or algorithmic innovation.
- Unrealistic Threat Model: It assumes attackers can fully control local data and observe global model updates continuously, which may not hold in real federated settings.
- Weak and Outdated Defenses: Evaluation against older baselines (e.g., Krum, Bulyan) limits the credibility of claimed robustness.
- Lack of Theoretical or Interpretive Insight: The paper does not analyze why or when the method works, nor provide visualization or formal justification for stealthiness.

**Questions:**

1. How realistic is the assumption that malicious clients can directly control and continuously optimize shifters during federated training, especially when secure aggregation or differential privacy is applied?
2. What happens if the global model is not shared in every round (e.g., in partial sharing or privacy-preserving FedGL)? Does the “Find” stage still work?
3. How sensitive is the performance to the number of malicious clients? Most results fix |CM| = 4 — what if only one attacker participates?
4. Is there any measurable trade-off between stealthiness and attack success when using more aggressive shifters?
5. Could the same effect be achieved with a simpler single-stage attack if initialization is carefully chosen?
6. Have the authors attempted to visualize the feature-space movement of poisoned graphs to justify the “shift toward decision boundary” claim?

---

### Official Review · Reviewer_Jq1e · 2025-10-30

**Soundness:** 2
**Presentation:** 2
**Contribution:** 1
**Rating:** 2
**Confidence:** 3

**Summary:**

This paper proposes FedShift, a novel two-stage “Hide and Find” distributed adversarial attack. It first embeds a hidden "shifter" during training to steer the model, and subsequently uses it as a basis for the rapid generation of adversarial examples. Experimental results show that the proposed method significantly outperforms existing techniques in terms of attack success rate, robustness against defenses, and efficiency

**Strengths:**

1. The method is innovative, introducing a "distributional shift" strategy in federated graph learning that enhances stealthiness.

2. The experimental evaluation is comprehensive, utilizing six large-scale datasets that span multiple real-world scenarios

**Weaknesses:**

1. Only the AAS metric is used. It lacks comparison with other commonly used attack evaluation metrics and does not provide a theoretical justification.

2. The evaluated defense methods do not appear to be state-of-the-art. Using simple practical defenses as a reference does not sufficiently demonstrate the true effectiveness of the attack. It is recommended that the authors survey relevant prior work, such as [1].

3. This paper lacks explicit discussion on preventing malicious use of the attack or proposing appropriate defensive measures.

[1] Nguyen, Thien Duc, et al. "{FLAME}: Taming backdoors in federated learning." 31st USENIX Security Symposium (USENIX Security 22). 2022.

Minor concern: Figure 1 does not explicitly show that the attacker poisons only part of the data, which could potentially mislead readers

**Questions:**

1. Are there any effective defenses or corresponding mitigation strategies against FedShift?

2.It is not very clear how the authors compare their method with GTA. As far as I understand, GTA targets GNNs rather than FedGL.

3.Adversarial attacks and backdoor attacks are distinct types of attacks. Adversarial attacks aim to cause incorrect predictions by adding small perturbations to inputs during the testing phase. However, the "Attacker’s objective" described in the paper seems to correspond to a backdoor attack. Could the authors have confused these concepts?

---

### Official Review · Reviewer_JZox · 2025-11-01

**Soundness:** 3
**Presentation:** 3
**Contribution:** 2
**Rating:** 4
**Confidence:** 3

**Summary:**

The paper proposes FedShift, a two-stage distributed adversarial attack on Federated Graph Learning (FedGL). Stage 1 injects "hidden shifters" that gently push poisoned graphs toward target class boundaries without crossing them. Stage 2 uses these shifters as initialization to efficiently find adversarial perturbations after federated training completes.

**Strengths:**

1.  The combination of backdoor and adversarial attack paradigms is creative, addressing limitations of each method when used independently.
2. Overall, the paper is well written and clearly explained.
3.  The method shows significant improvements in attack success rate and efficiency.

**Weaknesses:**

1. The assumption that attackers can "continuously optimize shifters throughout the whole federated process" is strong and may not reflect realistic scenarios.
2. The number of clusters k shows high sensitivity on some datasets (Figure 5).
3. No sensitivity analysis on loss weights.
4. No comparison with spectral defense techniques or pruning based defenses.

**Questions:**

1. Can you provide a mathematical characterization of how shifting graphs toward target class boundary without crossing boundary ensures stealthiness?
2. Can you elaborate more on the high sensitivity of cluster k on some datasets?
3. Are you considering both stage 1 and stage 2 for the computational comparison with other attacks?

---

### Note · Authors · 2025-11-19

I have read and agree with the venue's withdrawal policy on behalf of myself and my co-authors.